# Effect of Type and Dietary Fat Content on Rabbit Growing Performance and Nutrient Retention from 34 to 63 Days Old

**DOI:** 10.3390/ani11123389

**Published:** 2021-11-27

**Authors:** Saiz del Barrio Alejandro, García-Ruiz Ana Isabel, Nicodemus Nuria

**Affiliations:** 1Trouw Nutrition R&D Poultry Research Centre, 45950 Casarrubios del Monte, Spain; a.saiz.b@trouwnutrition.com (S.d.B.A.); ai.garcia@trouwnutrition.com (G.-R.A.I.); 2Departamento de Producción Agraria, ETSI Agrómica, Alimentaria y de Biosistemas, 28040 Madrid, Spain

**Keywords:** dietary fat, growing rabbit, performance, body composition, carcass composition, nutrient retention

## Abstract

**Simple Summary:**

An increase in the fat content of the diet increases the dietary energy concentration, and consequently, the growth and feed efficiency of the animals. The fatty acids (FA) profile of the fat source can also affect animal body composition. The purpose of this study was to test the effect of some fat sources added at different levels in growing rabbit feed. In this study, it was observed that the increment of dietary fat improved nitrogen efficiency utilization and reduced nitrogen excretion; the fat source also affected animal performance and mortality rate.

**Abstract:**

The study was carried out on individually and collectively housed growing rabbits from 34 to 63 days of age. Two experiments were conducted using three fat sources: Soybean oil (SBO), Soya Lecithin Oil (SLO), and Lard (L; Exp. 1), and SBO, Fish Oil (FO), and Palm kernel Oil (PKO; Exp. 2), added at two inclusion levels (1.5 and 4.0%). In both trials, 180 rabbits were housed in individual cages and additional 600 rabbits in collective cages from day 34 to 63. Animals fed with 4% dietary fat showed lower Daily Feed Intake (DFI) and Feed Conversion Ratio (FCR) than those fed with 1.5%, except in the individually housed animals in Exp. 1. In the collective housed group in Exp. 1, DFI was a 4.8% higher in animals fed with diets containing lard than those fed with SBO (*p* = 0.036). Lard inclusion also tended to reduce mortality (*p* = 0.067) by 60% and 25% compared with SBO and SLO diets, respectively. Mortality was the highest with the higher level of soya lecithin (14% vs. 1%, *p* < 0.01). A similar mortality rate was observed in the lowest level of SBO. In the grouped-housed animals in Exp. 2, a decrease of DFI (−12.4%), Bodyweight (BW) at 63 d (−4.8%), and Daily Weight Gain (DWG) (−7.8%) were observed with the inclusion of fish oil (*p* < 0.01) compared to other fat sources. Fish oil also tended to increase (*p* = 0.078) mortality (13.2%) compared with palm kernel oil (6.45%); similar results were found when animals were individually housed. The overall efficiency of N retention (NRE) increased with the highest level of fat in Exp. 1 (34.9 vs. 37.8%; *p* < 0.0001). It can be concluded that lard and palm kernel oil are alternative sources of fat due to the reduction of mortality. The inclusion of fish oil impaired animal productivity and increased mortality. An increment of the dietary fat level improved FCR and overall protein retention efficiency.

## 1. Introduction

The use of fatty ingredients in growing rabbit feed was widely studied [1]. Fat addition in rabbit diets increases digestible energy and improves feed efficiency [2,3,4]. When fat addition is low or moderate (2–6%), reduction of feed intake is observed and nutrient digestibility and feed efficiency may improve [3,5]. However, the fat inclusion level is usually limited, between 2 and 4%, due to negative effects in feed pelleting [4,6]. Young animals have a high-fat digestive capacity [7] as high lipase activity has been observed in 15 d old rabbits [8]. The effect of fat source and the inclusion level on nutrient retention is related to the fat digestibility and depends on the fatty acid profile; fat digestibility increases when the level of unsaturated fatty acids is greater [3,4,9]. In other studies, the efficiency of nitrogen retention has been improved by increasing fat addition [10] or increasing w-3 fatty acid content [11]. Moreover, fatter carcasses were also found when dietary fat content is increased [1,12].

On the other hand, it is known that short-chain fatty acids, like caprylic and capric acids, have a positive effect on rabbit gut health. In a previous study [13], mortality was reduced in rabbits fed 1% caprylic and capric acids compared with the control group (15.7 vs. 27.8%). Likewise, w-3 fatty acids seem to be involved in immune response development [14,15]. Maertens et al. [16] reported an increase in the post-weaning viability of young rabbits fed a diet with a high w-3/w-6 ratio. These types of fatty acids could lead to a well-balanced digestive tract maturation and the development of a better immune system, preventing enteropathy incidence and limiting the use of antibiotics [16]. This study aimed to determine the effect of the fat level and source on growth performance, mortality, nutrient retention, and carcass composition of 34 to 63 day-old rabbits.

## 2. Materials and Methods

This study was approved by the Ethics Committee of Trouw Nutrition R&D Poultry Research Centre (ethic code 13-2019 provided by the Livestock Service of Castilla-La Mancha, Spain, as authorized agency). Rabbits were handled according to the principles for the care of experimental animals [17].

### 2.1. Animal and Housing

This study was carried out at the Trouw Nutrition Poultry Research Centre, located in Casarrubios Del Monte (Toledo, Spain) with the collaboration of the Universidad Politécnica de Madrid. Two consecutive flocks of seven hundred eighty New Zealand × Californian rabbits each were used in two experiments (Exp. 1 and Exp. 2). Animals were weaned 34 d after birth. In each experiment, 180 rabbits were individually housed in flat-deck cages (250 × 440 × 300 mm). Animals were blocked per litter, assigning animals from the same litter to different treatments.

To estimate better the effect of treatments on mortality in each of the experiments, 600 extra animals were allocated in collective cages (5 animals per cage) in a different room. Two animals from the same litter were not allocated in the same cage. Animals were fed until day 63 and kept under controlled environmental conditions (room temperature between 16 and 24 °C, with a light: dark cycle of 8–16 h; the light was switched on at 07.30 h). At the end of the study, all animals were sent to a commercial slaughterhouse (day 63).

### 2.2. Diets

All diets were formulated to meet or exceed rabbit nutrient requirements according to the CVB [18]. In Exp. 1, three diets were arranged in a 3 × 2 factorial arrangement with fat source [Soybean oil (SBO), Soya Lecithin Oil (SLO), and Lard (L)] and dietary fat content (1.5 and 4%) as the main factors. Exp. 2 was also arranged as a 3 × 2 factorial design but using SBO, Fish Oil (FO), and Palm kernel Oil (PKO) as fat sources and added at the same levels as in Exp. 1. In both experiments, 20 collective and 30 individual cages were used; each cage was considered a treatment replicate. In Table 1 and Table 2, the ingredient and analyzed chemical composition of the diets used in each experiment are shown. The calculated amino acids profile of each diet is also shown. In Table 3, the analyzed fatty acids profile of each fat source is presented.

### 2.3. Growth Performance Study

In both experiments and housing systems, the rabbit weight (BW) and feed intake were registered at weaning (34 d) and on 49 and 63 d. Daily Feed Intake (DFI), Daily Weight Gain (DWG), and Feed Conversion Ratio (FCR) were calculated from 34 to 49 d, 49 to 63 d, and 34 to 63 d. Dead animals were recorded daily, and mortality was calculated per period relative to the initial amount of animals at the start of each period.

### 2.4. Body and Carcass Composition Measurement

In both experiments, whole body and carcass composition [water (%), protein (%DM), fat (%DM), ash (%DM), and energy (kJ/100 g DM)] were estimated on days 34 and 63 in 60 individually housed rabbits (10 animals per treatment) using the Bioelectrical Impedance (BIA) method and the prediction equations developed by Saiz et al. [20,21]. BIA measurements were performed twice on each animal and with a 30 min interval between readings. BIA measurements were performed between 11.00 and 13.00 h and using a four-terminal body composition analyzer (Model BIA-101, RJL Systems, Detroit, MI, USA).

### 2.5. Energy and Nitrogen Carcass Retention and Excretion

Using the body and carcass composition BIA data, total energy (kcal) and nitrogen (protein/6.25, g) carcass content were estimated and used to calculate the daily energy and nitrogen retention (ER, kcal/d and NR, g/d respectively) between 34 and 63 d. Moreover, daily digestible nitrogen and digestible energy intake (DNi, g/d and DEi, kcal/d, respectively) were also estimated. The overall nitrogen retention efficiency (NRE, %) and overall energy retention efficiency (ERE, %) in the carcass were calculated as follow:NRE, % = 100 × (NR/DNi)
ERE, % = 100 × (ER/DEi)

Total nitrogen and energy excretion as skin and organs, feces, or heat production and urine were calculated as follow:-Nitrogen retained on skin and organs (g/d): g N retained in the whole body—g N retained on the carcass.-Nitrogen excreted on faces (g/d): N total intake—DNi.-Nitrogen excreted on urine (g/d): DNi—g N retained on the carcass—N excreted on skin and organs.-Energy retained on skin and organs (kcal): GE retained in the whole body—GE retained in the carcass.-Energy excreted on feces (kcal/d): GEi—DEi.-Energy excreted on urine and used for heat production (kcal/d): Dei—GE retained in carcass—GE excreted in skin and organs.

### 2.6. Chemical Analysis

Diets were analyzed following the AOAC methods [22]: DM (934.01), CP (Dumas Method, N × 6.25; 968.06), Ash (942.05), and fat (RD 609/1999 n°4, previous acid hydrolysis [23]). The GE content was determined using an adiabatic calorimetric bomb (Model 6100, Parr Instrument Company, Moline, IL, USA). The DE and DN content of the diets were estimated using each ingredient digestibility coefficient as described by Maertens et al. [19]. The fatty acid profiles of the fat sources were determined by Gas Chromatography, using a gas chromatograph (model HP 6890, Hewlett Packard, Palo Alto, CA, USA), with He as transporting gas. The methylation was done with MeONa-trifluoride.

### 2.7. Statistical Analysis

A variance analysis was performed for growth traits, body, and carcass chemical composition, and energy and nitrogen balances in both experiments by using the GLM procedure of SAS. Within each housing system (individual and collective), data were analyzed as a factorial arrangement using the litter as a blocking effect and the fat source and level as the main sources of variation. Weaning weight was used as a linear covariate when analyzing performance data. Mortality was analyzed as a binomial distribution using the logit transformation of the GLIMMIX procedure of SAS. A Tukey’s test was used for mean comparisons. For growth performance analysis, the cage (individual or collective) was considered as the experimental unit, while for body and carcass composition and energy and nitrogen carcass retention and excretion, the animal was considered the experimental unit.

## 3. Results

### 3.1. Growth Performance

As individually housed rabbits tend to have lower mortality rates than rabbits collectively housed, the same study design was performed on individual and collective cages to check the effect of treatment on mortality rates.

#### 3.1.1. Experiment 1

The effect of fat source, fat level inclusion, and their interaction on individually housed rabbit performance and mortality from 34 to 63 d are presented in Table 4. The fat source effects and the interaction between fat source and level were not significant in most of the parameters recorded, independently of the studied period. A significant effect of the fat level was found on DFI in the first growing period (34–49 d; *p* = 0.02), on FCR in the second (49–63 d; *p* = 0.001), and in the whole experimental period (34–63 d; *p* < 0.0001). Rabbits fed the lowest fat level (1.5%) showed 4.9, 6.8, and 5.2% higher DFI34–49, FCR49–63, and FCR34–63 values, respectively, than rabbits fed the highest level (4.0%).

Performance and mortality results of rabbits collectively housed from 34 to 63 d of age are shown in Table 5. Rabbits fed with soybean oil showed significantly lower DFI (*p* = 0.036) and tended to have a higher mortality rate (*p* = 0.067) than rabbits fed the diets containing lard (105 vs. 110 g/d and 10.0 vs. 3.97%, respectively). Rabbits fed the diets with lecithin (SLO) showed intermediate values for both parameters, although an interaction (fat source × fat level) was observed on mortality for diet SLO because rabbits fed with this source had the highest and the lowest mortality rate (14.0 vs. 1.01% for diets containing 4 and 1.5% of fat, respectively).

A significant effect of the fat inclusion level (1.5 vs. 4.0%) was detected on DFI and FCR. Rabbits fed the lowest fat level (1.5%) showed a 6.0, and 2.97% higher DFI and FCR, respectively, than rabbits fed the highest level (4.0%). With the lowest level of fat, BW at 63 d and DWG tended to be 2.1 and 3.4% higher than with the highest, respectively.

#### 3.1.2. Experiment 2

Rabbits housed individually and fed the FO diet from 34–49 d showed, on average, 4.0, 8.3, and 9.1% lower BW49, DWG, and DFI, respectively, than rabbits fed the SBO and PKO diets (*p* < 0.0001; Table 6). Likewise, between 49–63 d the rabbits fed the FO diet showed, on average, a 3.2, 5.1, and 3.9% lower BW63, DFI, and FCR, respectively, than the other two fat sources. During the total growing period, the same tendency was observed, the FO diet impaired DWG (4.98%; *p* = 0.001) and DFI (6.72%; *p* < 0.0001) but tended to improve FCR (2.21%; *p* = 0.086) compared with SBO and PKO diets.

Regarding the fat level effect, rabbits fed with the highest fat content reduced DFI by 4.1% from 34–49 d (*p* = 0.0018), 3.5% from 49–63 d (*p* = 0.0062), and 3.8% (*p* = 0.0008) from 34–63 d compared with the lowest fat content. An interaction was found between fat source and level in DFI, from 34–49 d (*p* = 0.018) and in the total growing period (*p* = 0.098) because rabbits fed with the highest FO level had the lowest DFI (*p* < 0.05), while animals fed with SBO had similar feed intake between the two levels. As DFI was decreased without impairing DWG, FCR was better for the highest level of fat in the second growing period (4.7%; *p* = 0.0013) and the whole period (2.70%; *p* = 0.0045).

When animals were housed collectively similar effects were observed (Table 7). Decreases in DFI (12.4%), BW at 63 d (4.8%), and DWG (7.8%) were observed with the inclusion of FO compared with the other two diets (*p* < 0.01). These last two traits were impaired by the highest level of fish oil (5.6 and 9.5%, respectively, (*p* < 0.01)). The inclusion of fish oil also tended to increase (*p* = 0.078) mortality (13.2%) compared with palm kernel oil (6.45%); mortality with SBO was intermediate (8.10%).

In both experiments, mortality rates in individually housed animals were close to 0 for all treatments, therefore values are not shown in the tables.

### 3.2. Body and Carcass Composition and Nutrient Retention

Differences in the estimated whole body and carcass composition with different fat sources, inclusion levels, and their interaction were not detected (Table 8 and Table 9).

### 3.3. Digestible Energy and Digestible Protein Carcass Retention

Increasing the fat content led to a decrease of the digestible nitrogen intake (DNi) (1.83 vs. 1.92 g/d; *p* = 0.068 in Exp. 1 and 1.79 vs. 1.95 g/d; *p* = 0.014 in Exp. 2) (Table 10 and Table 11). As the nitrogen retained (NR) in the carcass was similar for both fat levels (0.68 g/d (Exp. 1) and 0.69 g/d (Exp. 2)), the overall N retention efficiency (NRE) increased with the highest level of fat, but it was only significancy in Exp. 1 (34.9 vs. 37.8%; *p* < 0.0001); in Exp. 2 a tendency was found (36.2 vs. 38.0% in Exp. 2; *p* < 0.064).

In both experiments, nitrogen excretion in feces was lower in animals fed with the highest level of fat (0.782 vs. 0.868 g/d; *p* = 0.0001 in Exp. 1 and 0.745 vs. 0.865 g/d; *p* < 0.0001 in Exp. 2). The same effect was detected for nitrogen excreted as urine (0.702 vs. 0.822 g/d; *p* < 0.0001 in Exp. 1 and 0.694 vs. 0.799 g/d; *p* = 0.014 in Exp. 2) and energy excreted in feces (142 vs. 156 kcal/d; *p* = 0.0004 in Exp. 1 and 144 vs. 154 g/d; *p* = 0.050 in Exp. 2). In Exp. 1, energy excreted as urine and heat production was significantly higher for the highest level of dietary fat (216 vs. 204 kcal/d; *p* < 0.017). No effect of fat source, level or the interaction was found on ERE levels in either experiment.

## 4. Discussion

### 4.1. Growth Performance

The average weight and feed intake of the group housed animals were 11% and 25%, respectively, lower than those of the individually housed animals, due to the greater competition to accessing the feeder among animals in the cage. However, FCR was similar between both of them. The results obtained were similar between animals who grew in individual and collective cages, but the differences between treatments were greater in animals housed collectively, because of the higher number of animals used. In all the cases, animals fed with diets higher in fat content needed a lower feed intake to meet their energy requirements. As the level of fat did not show any effect on weight gain, animals fed with the highest level of fat had an improved feed conversion ratio. These results are in agreement with other studies [2,3] showing that an increment in the fat level, over 2–3%, leads to a decrease in feed intake and an improvement in feed efficiency.

In Exp. 1, the grouped-animals feed intake was higher, and mortality tended to be lower in animals fed lard than in animals fed the soybean oil supplemented diets. These results are not in agreement with another work [24] that observed an increase in the mortality rate of rabbits fed a diet with 3% lard addition (45%) in diets supplemented with vegetable oils (linseed (34%) or sunflower oil (37%). The mortality in our study was lower than in the other works (between 1.01 and 14.0%) and the highest value was reached with the 4% lecithin diet. Other authors [3] did not find significant differences between diets including lard or lecithin, either at 3 or 6% inclusion levels, with a mortality rate of 3.5% on average.

In Exp. 2, a negative effect on feed intake was found with the inclusion of fish oil. The decrease in feed intake could be due to lower palatability derived from rancidity issues. The tendency to increase the grouped animal mortality with the use of fish oil was unexpected as the positive effect of these fatty acids on the immune status of rabbits and humans is well known [14,15,24]. Indeed, Rodriguez et al. [25] found improvement in the morbidity rate with the inclusion of fish oil; however, Delgado et al. [11] did not observe impairment on the mortality rates with a low w-6/w-3 ratio. Similarly, Kelley et al. [26] observed that the rabbit immune system was less sensitive to fish oil compared with other vegetable omega-3 sources as linseed oils. The explanation may be based on the length difference of the carbon chain. On the contrary in the present study, palm kernel oil had a positive effect on mortality rate compared to soybean oil. Palm kernel oil is rich in medium and short-chain fatty acids, including caprylic and capric acids that are known to improve rabbit health [13]). On the other hand, lard and palm kernel oil have higher levels of w-6/w-3 ratio than soybean oil, fish oil, or soybean lecithin, and the results obtained in this study disagree with those found elsewhere [16]. Maertens et al. [16] reported lower mortalities in diets rich in w-3 fatty acids, probably related to immune system regulation with these fatty acids. As palm kernel oil and lard including treatments showed similar performance results, but lower mortality rates than soybean oil, both fat sources could be considered as potential alternatives for replacing soybean oil.

### 4.2. Body and Carcass Composition and Nutrient Retention

BIA-derived whole body and carcass composition were similar in both experiments and showed some differences with the values observed in previous studies that determined the whole empty body composition when using the comparative slaughter method. Values found in these studies were: [27]: 50.3% (NRE) and 18.9% (ERE); [28]: 41.0% (NRE) and 25.0% (ERE); [29]: 51.5% (NRE) and 21.2% (ERE); and [30]: 37.7% (NRE) and 25.8% (ERE). However, in more recent studies, the BIA method was used to determine nutrient retention ([11]: 36.5% (NRE) and 18.6% (ERE), and [31]: 32% (NRE) and 20% (ERE)). In our work, the values were on average 36.3 and 37.1% (NRE for Exp. 1 and 2), and 18.8 and 20.2% (ERE for Exp. 1 and 2) and they agreed with those obtained in the BIA method works. The differences in ERE and NRE values among studies could be due to the different methodology used and the animals’ processing method because the skin is included in the empty body and carcasses, the skin is removed. 

Nitrogen excretion (2.00 g/d of N retained in skin and organs and excreted in feces and urine) and energy excretion (387 kcal/d retained in skin and organs and excreted on feces and urine and heat production) were similar to those obtained by Delgado et al. [11] (2.01 g/d, and 327 kcal/d excreted nitrogen and energy, respectively). Crespo et al. [31] also observed similar values of excretion (2.06 g/d and 382 kcal/d of nitrogen and energy excreted, respectively) in animals fed *ad libitum*.

Growing rabbits adjust feed intake depending on the energy content of the diet, to meet its energy requirements [32]. In our work, animals fed diets containing 4% added fat and higher DE content showed lower DFI; this response to dietary energy density has also been observed in previous studies [33,34]. However, as the digestible nitrogen content among diets was similar, independently of the dietary fat level, the digestible protein intake of those animals was reduced. When animals lack or are restricted in a specific nutrient, they become more efficient in nutrient absorption and retention [35], as was observed in NRE in this study. These animals also had lower nitrogen excretion in skin and organs, and feces and urine. This result agrees with Parigi Bini et al. [36] and Fernández and Fraga [30] that also observed an improvement in NRE with increased dietary fat content. Due to the same DE ingestion and retention, ERE was not affected by the treatments despite those animals excreted less energy in feces when the dietary fat content was higher, as a consequence of a higher, but not significant, energy retention in the carcass.

## 5. Conclusions

From the results obtained in this work, it can be concluded that: (1) the dietary fat increment from 1.5 to 4% improved nitrogen efficiency and reduced nitrogen excretion; (2) lard and palm kernel oil were a better choice than soybean oil as both reduced mortality without impairing performance; (3) fish oil, with the highest w-3/w-6 ratio, is not a suitable fat source choice to replace soybean oil as its tended to increase rabbit mortality; (4) whole body and carcass composition were not affected by dietary fat source either at 1.5 or 4% inclusion levels.

## Figures and Tables

**Table 1 animals-11-03389-t001:** Ingredients and chemical composition of the diets used in Experiment 1.

Source ^1^	SBO	SBO	SLO	SLO	L	L
Inclusion level, %	1.5	4.0	1.5	4.0	1.5	4.0
Ingredients composition, %						
Wheat bran	30.0	29.2	30.0	29.2	30.0	29.2
Barley	20.4	19.8	20.4	19.8	20.4	19.8
Sunflower meal	10.6	10.3	10.6	10.3	10.6	10.3
Alfalfa	13.0	12.7	13.0	12.7	13.0	12.7
Wheat straw	5.00	4.90	5.0	4.90	5.0	4.90
Sugar beet pulp	15.0	14.6	15.0	14.6	15.0	14.6
Soybean oil	1.50	4.00	1.05	2.80	-	-
Soybean lecithin oil	-	-	0.45	1.20	-	-
Lard	-	-	-	-	1.50	4.00
Vitamin and Mineral Premix ^2^	0.50	0.50	0.50	0.50	0.50	0.50
Salt	0.50	0.50	0.50	0.50	0.50	0.50
CaCO_3_	1.25	1.20	1.25	1.20	1.25	1.20
Sepiolite	2.00	2.00	2.00	2.00	2.00	2.00
L-Lysine	0.18	0.20	0.18	0.20	0.18	0.20
L-Threonine	0.07	0.10	0.07	0.10	0.07	0.10
Total	100.0	100.0	100.0	100.0	100.0	100.0
Theoretical amino acids composition, %						
Lysine	0.740	0.740	0.740	0.740	0.740	0.740
Metionine	0.254	0.254	0.254	0.254	0.254	0.254
TSAA	0.517	0.517	0.517	0.517	0.517	0.517
Threonine	0.600	0.600	0.600	0.600	0.600	0.600
Analyzed chemical composition, %						
Dry Matter	89.9	90.5	90.6	90.6	90.6	90.8
Crude Protein	14.2	13.7	14.3	13.8	14.2	13.7
Neutral Detergent Fiber	34.6	34.7	35.6	34.8	34.9	34.6
Acid Detergent Fiber	18.5	18.4	18.3	18.3	19.1	18.4
Acid Detergent Lignin	4.15	4.07	3.96	4.40	4.68	4.36
Starch enzymatic	15.6	14.3	15.4	13.6	13.8	14.3
Fat	3.55	5.85	3.50	5.40	3.50	5.85
Ash	8.87	8.76	9.20	9.06	9.32	9.00
Gross Energy, kcal/kg	3720	3861	3734	3831	3727	3863
Digestible Energy, kcal/kg ^3^	2463	2677	2453	2649	2455	2655
Digestible Nitrogen ^3^	1.57	1.54	1.57	1.54	1.57	1.54

^1^ SBO: Soybean Oil; SLO: Soya Lecithin oil; L: Lard; TSAA: Total Sulphur Aminoacids; ^2^ Provided by Trouw Nutrition España. (Tres Cantos, Spain). Mineral and vitamin composition (per kg of complete diet): 240 mg of S; 240 mg of Mg as MgO; 20 mg of Mn as MnO; 75 mg of Zn as ZnO; 180 mg of Cu as CuSO_4_·5H_2_O; 1.1 mg of I as KI; 0.5 mg of Co as CoCO_3_·H_2_O; 0.06 mg of Se as SeO_2_; 7.8 mg of Fe as FeCO_3_; 12,000 UI of Vitamin A; 10,800 UI of Vitamin D3; 45 mg of Vitamin E dl-alfa-tocopherol acetate; 1.2 mg of Vitamin K; 2 mg of Vitamin B1; 60 mg of Vitamin B2; 2 mg of Vitamin B6; 10 mg of Vitamin B12; 40 mg of Niacin; 20 mg of calcium pantothenate; 18.4 mg of pantothenic acid; 5 mg of folic acid; 75 mcg of biotin; 260 mg of choline chloride; 10 mg of Diclazuril 0.5 g/100 g (Clinacox 0.5% Premix ©, Elanco, Greenfield, USA); 0.12 mg of Butohidroxianisol; 13.2 mg of Butohidroxitolueno; 38.4 mg of ethoxyquin. ^3^ Estimated by using the digestibility coefficients of Maertens et al. (2002) [19].

**Table 2 animals-11-03389-t002:** Ingredients and chemical composition of the diets used in Experiment 2.

Source ^1^	SBO	SBO	PKO	PKO	FO	FO
Inclusion level, %	1.5	4.0	1.5	4.0	1.5	4.0
Ingredients composition, %						
Wheat bran	23.1	22.5	23.1	22.5	23.1	22.5
Barley	27.6	26.9	27.6	26.9	27.6	26.9
Sunflower meal	13.6	13.3	13.6	13.3	13.6	13.3
Alfalfa	13.4	13.1	13.4	13.1	13.4	13.1
Wheat straw	5.00	4.84	5.00	4.84	5.00	4.84
Sugar beet pulp	11.0	10.7	11.0	10.7	11.0	10.7
Soybean oil	1.50	4.00	-	-	-	-
Fish oil	-	-	1.50	4.00	-	-
Palm kernel oil	-	-	-	-	1.50	4.00
Vitamin and Mineral Premix ^2^	0.50	0.49	0.50	0.49	0.50	0.49
Salt	0.50	0.49	0.50	0.49	0.50	0.49
CaCO_3_	1.50	1.46	1.50	1.46	1.50	1.46
Sepiolite	2.03	1.95	2.03	1.95	2.03	1.95
L-Lysine	0.20	0.20	0.20	0.20	0.20	0.20
L-Threonine	0.07	0.07	0.07	0.07	0.07	0.07
Total	100.0	100.0	100.0	100.0	100.0	100.0
Theoretical amino acids composition, %						
Lysine	0.740	0.722	0.740	0.722	0.740	0.722
Metionine	0.262	0.256	0.262	0.256	0.262	0.256
TSAA	0.527	0.514	0.527	0.514	0.527	0.514
Threonine	0.600	0.585	0.600	0.585	0.600	0.585
Analyzed chemical composition, %						
Dry Matter	91.5	91.9	92.1	92.2	91.9	91.3
Crude Protein	14.7	14.0	14.6	14.0	14.7	14.1
Neutral Detergent Fiber	32.2	32.5	33.1	33.0	32.9	31.3
Acid Detergent Fiber	17.9	17.5	18.0	17.8	17.6	17.0
Acid Detergent Lignin	5.01	4.97	4.95	5.38	4.85	4.53
Starch enzymatic	16.1	15.4	15.9	15.4	15.8	15.7
Fat	3.00	5.30	3.30	5.70	3.20	5.30
Ash	9.50	9.50	9.60	9.40	9.70	9.20
Gross Energy, kcal/kg	3738	3864	3773	3901	3757	3853
Digestible Energy, kcal/kg ^3^	2453	2601	2453	2601	2453	2601
Digestible Nitrogen ^3^	1.61	1.58	1.61	1.58	1.61	1.58

^1^ SBO: Soybean Oil; FO: Fish Oil; PKO: Palm kernel oil; TSAA: Total Sulphur Aminoacids. ^2^ Provided by Trouw Nutrition España. (Tres Cantos, Spain). Mineral and vitamin composition (per kg of complete diet): 240 mg of S; 240 mg of Mg as MgO; 20 mg of Mn as MnO; 75 mg of Zn as ZnO; 180 mg of Cu as CuSO_4_·5H_2_O; 1.1 mg of I as KI; 0.5 mg of Co as CoCO_3_·H_2_O; 0.06 mg of Se as SeO_2_; 7.8 mg of Fe as FeCO_3_; 12,000 UI of Vitamin A; 10,800 UI of Vitamin D3; 45 mg of Vitamin E dl-alfa-tocopherol acetate; 1.2 mg of Vitamin K; 2 mg of Vitamin B1; 60 mg of Vitamin B2; 2 mg of Vitamin B6; 10 mg of Vitamin B12; 40 mg of Niacin; 20 mg of calcium pantothenate; 18.4 mg of pantothenic acid; 5 mg of folic acid; 75 mcg of biotin; 260 mg of choline chloride; 10 mg of Diclazuril 0.5 g/100 g (Clinacox 0.5% Premix ©, Elanco, Greenfield, USA); 0.12 mg of Butohidroxianisol; 13.2 mg of Butohidroxitolueno; 38.4 mg of ethoxyquin. ^3^ Estimated by using the digestibility coefficients of Maertens et al. (2002) [19].

**Table 3 animals-11-03389-t003:** Analyzed fatty acid profile of the fat sources used in the experimental diets.

Source ^1^		SBO	SLO	L	FO	PKO
Fatty acid profile (%)						
	C < 14	<0.2	<0.2	<0.2	0.36	64.5
Miristic	C14:0	<0.2	<0.2	1.10	8.60	17.1
Palmític	C16:0	10.9	16.6	21.0	20.3	5.70
Palmitoleic	C16:1	<0.2	<0.2	<0.2	8.40	-
Estearic	C18:0	3.80	3.60	10.5	3.10	0.80
Oleic	C18:1	23.8	18.8	52.7	25.0	7.60
Linoleic	C18:2	53.5	53.5	8.30	18.5	1.30
Linolenic	C18:3	6.20	5.50	0.60	2.70	0.05
	C > 20	0.70	1.70	2.60	11.7	3.00
Unsaturated/Saturated		5.30	3.60	2.00	1.70	0.10
Total w-3		6.20	5.50	0.60	7.04	0.05
Total w-6		53.5	53.5	8.80	18.8	1.30
w-6/w-3		8.60	10.0	14.3	2.67	25.0

^1^ SBO: Soybean Oil; SLO: Soya Lecithins oil; L: Lard; FO: Fish Oil; PKO: Palm kernel oil.

**Table 4 animals-11-03389-t004:** Effect of dietary fat source and fat inclusion level on rabbit performance from 34 to 63 days (experiment 1, individually housed).

	Fat Source ^1^			Fat Level, %		rsd ^2^	*p_s_*	*p_l_*	*p_s x l_*
	SBO	SLO	L	1.5	4.0				
N	60	60	60	90	90				
34–49 days									
BW 34 d, g	742	759	766	754	757	-	-	-	-
BW 49 d, g	1494	1521	1498	1510	1499	125	0.44	0.56	0.11
DWG, g/d	49.2	51.0	49.4	50.2	49.5	8.3	0.44	0.56	0.11
DFI, g/d	103	106	106	108	103	14	0.48	0.02	0.21
FCR	2.13	2.07	1.91	2.13	1.94	1.15	0.57	0.24	0.32
49–63 days									
BW 63 d, g	2122	2151	2145	2121	2158	170	0.64	0.15	0.42
DWG, g/d	45.4	44.7	44.8	43.9	46.0	8.4	0.89	0.11	0.50
DFI, g/d	138	139	136	138	137	22	0.65	0.63	0.26
FCR	3.07	3.20	3.08	3.22	3.01	0.38	0.16	0.001	0.60
34–63 days									
DWG, g/d	47.1	48.1	47.9	47.1	48.3	5.9	0.64	0.15	0.42
DFI, g/d	119	123	121	123	120	15	0.50	0.19	0.36
FCR	2.54	2.55	2.53	2.61	2.48	0.14	0.72	<0.0001	0.73

^1^ SBO: Soybean oil; SLO: Soya Lecithin oil; L: Lard. ^2^ rsd: residual standard deviation. *p_s_*: probability of the source; *p_l_*: probability of the level; *p_s x l_*: probability of the interaction source x level.

**Table 5 animals-11-03389-t005:** Effect of dietary fat source and fat inclusion level on rabbit performance and mortality from 34 to 63 days (experiment 1, collectively housed: 5 animals/cage).

Fat Source ^1^	SBO		SLO		L		Fat Source			Fat Level, %		rsd ^2^	*p_s_*	*p_l_*	*p_s x l_*
Fat Level	1.5	4.0	1.5	4.0	1.5	4.0	SBO	SLO	L	1.5	4				
N	20	20	20	20	20	20	40	40	40	60	60				
BW 34 d, g	724	734	731	733	733	733	729	732	733	730	733	-	-	-	-
BW 63 d, g	1944	1875	1925	1912	1967	1931	1910	1919	1949	1946	1906	114	0.28	0.063	0.55
DWG, g/d	44.9	42.4	44.2	43.7	45.8	44.4	43.6	44.0	45.1	45.0	43.5	4.2	0.28	0.063	0.55
DFI, g/d	108	102	109	104	114	106	105 ^b^	106 ^ab^	110 ^a^	110	104	8	0.036	0.0003	0.72
FCR	2.42	2.40	2.48	2.39	2.49	2.39	2.41	2.43	2.44	2.46	2.39	0.12	0.58	0.001	0.22
Mortality, %	12.1 ^a^	7.96 ^ab^	1.01 ^b^	14.0 ^a^	2.96 ^b^	4.97 ^b^	10.0	7.49	3.97	5.36	8.97	12.0	0.067	0.16	0.006

^1^ SBO: Soybean oil; SLO: Soya Lecithin oil; L: Lard; ^2^ rsd: residual standard deviation Experimental unit is the cage (5 rabbits per cage). Means in the same row with different letters show significant differences (*p <* 0.05) among treatments. *p_s_*: probability of the source; *p_l_*: probability of the level; *p_s x l_*: probability of the interaction source x level.

**Table 6 animals-11-03389-t006:** Effect of dietary fat source and fat inclusion level on rabbit performance from 34 to 63 days (experiment 2, individually housed).

Fat Source ^1^	SBO		FO		PKO		Fat Source			Fat Level, %		rsd ^2^	*p_s_*	*p_l_*	*p_s x l_*
Fat Level	1.5	4.0	1.5	4.0	1.5	4.0	SBO	FO	PKO	1.5	4				
N	30	30	30	30	30	30	60	60	60	90	90				
34–49 days															
BW 34 d, g	772	767	769	773	749	764	770	771	756	763	768	-	-	-	-
BW 49 d, g	1481	1485	1441	1398	1482	1464	1483 ^a^	1419 ^b^	1473 ^a^	1468	1449	77	<0.0001	0.10	0.24
DWG, g/d	51.1	51.4	48.3	45.2	51.2	49.9	51.3 ^a^	46.7 ^b^	50.6 ^a^	50.2	48.9	5.5	<0.0001	0.10	0.24
DFI, g/d	98.5 ^a^	98.3 ^ab^	94.1 ^b^	85.2 ^c^	100 ^a^	97.3 ^ab^	98.4 ^a^	89.6 ^b^	98.9 ^a^	97.6	93.6	1.5	<0.0001	0.0018	0.018
FCR	1.92	1.93	1.95	1.90	1.96	1.97	1.93	1.92	1.97	1.94	1.94	0.20	0.44	0.82	0.65
49–63 days															
BW 63 d, g	2139	2151	2094	2050	2137	2134	2146 ^a^	2072 ^b^	2136 ^a^	2124	2112	111	0.001	0.51	0.391
DWG, g/d	47.1	47.9	46.7	46.7	46.8	47.7	47.5	47.3	46.7	46.9	47.4	5.3	0.68	0.50	0.876
DFI, g/d	140	139	135	128	142	135	139 ^a^	132 ^b^	139 ^a^	139	134	12	0.0013	0.0062	0.260
FCR	3.00	2.96	2.91	2.77	3.05	2.85	2.96 ^a^	2.84 ^b^	2.94 ^a^	2.99	2.85	0.28	0.035	0.0013	0.436
34–63 days															
DWG, g/d	49.1	49.5	47.5	45.9	49.0	48.9	49.3 ^a^	46.7 ^b^	49.0 ^a^	48.5	48.1	4.1	0.001	0.508	0.392
DFI, g/d	119	118	115	107	121	116	119 ^a^	111 ^b^	119 ^a^	118	114	9	<0.0001	0.0008	0.098
FCR	2.43	2.40	2.41	2.33	2.48	2.38	2.41	2.37	2.43	2.44	2.37	0.15	0.086	0.0045	0.364

^1^ SBO: Soybean oil; FO: Fish oil; PKO: Palm kernel oil. ^2^ rsd: residual standard deviation. Means in the same row with different letters show significant differences (*p <* 0.05) among treatments. *p_s_*: probability of the source; *p_l_*: probability of the level; *p_s x l_*: probability of the interaction source x level.

**Table 7 animals-11-03389-t007:** Effect of dietary fat source and fat inclusion level on rabbit performance and mortality from 34 to 63 days (experiment 2, collectively housed: 5 animals/cage).

Fat Source ^1^	SBO		FO		PKO		Fat Source			Fat Level, %		rsd ^2^	*p_s_*	*p_l_*	*p_s x l_*
Fat Level	1.5	4.0	1.5	4.0	1.5	4.0	SBO	FO	PKO	1.5	4				
N	20	20	20	20	20	20	40	40	40	60	60				
BW 34 d, g	736	698	723	726	723	715	717	724	719	727	713	-	-	-	-
BW 63 d, g	1858	1875	1839	1735	1882	1896	1867 ^a^	1787 ^b^	1889 ^a^	1860	1835	130	0.0019	0.31	0.065
DWG, g/d	40.6	41.3	40.0	36.2	41.5	42.0	41.0 ^a^	38.1 ^b^	41.7 ^a^	40.7	39.8	1.0	0.0019	0.31	0.066
DFI, g/d	98.4	95.6	92.9	83.8	104	99.9	97.0 ^a^	88.4 ^b^	104 ^a^	98.4	93.1	4.5	<0.0001	0.012	0.40
FCR	2.42	2.32	2.33	2.33	2.51	2.39	2.37 ^ab^	2.33 ^a^	2.45 ^b^	2.42	2.35	0.18	0.018	0.034	0.36
Mortality, %	8.98	7.23	12.9	13.4	8.87	4.02	8.10	13.2	6.45	10.2	8.23	12.0	0.078	0.42	0.67

^1^ SBO: Soybean oil; FO: Fish oil; PKO: Palm kernel oil. ^2^ rsd: residual standard deviation. The experimental unit is the cage (5 rabbits per cage). Means in the same row with different letters show significant differences (*p <* 0.05) among treatments. *p_s_*: probability of the source; *p_l_*: probability of the level; *p_s x l_*: probability of the interaction source x level.

**Table 8 animals-11-03389-t008:** Body and carcass chemical composition of 63 d old rabbits estimated by bioelectrical impedance (BIA) (experiment 1, individually housed) (N = 10).

		Whole Body Composition					Carcass Composition				
Fat ^1^ Source	Fat Level %	Water %	Protein %DM	Ash %DM	Fat %DM	Energy kJ/100 g DM	Water %	Protein %DM	Ash %DM	Fat %DM	Energy kJ/100 g DM
SBO	1.5	69.2	51.8	10.6	31.1	2363	65.0	58.2	13.8	30.4	2350
	4.0	68.3	50.6	10.1	33.5	2422	64.6	57.4	13.6	31.7	2375
SLO	1.5	68.8	51.3	10.4	32.1	2392	64.9	58.1	13.8	31.0	2355
	4.0	68.5	51.1	10.3	32.4	2395	64.2	57.6	12.8	31.1	2410
L	1.5	68.3	50.8	10.1	33.4	2421	64.4	57.7	13.3	31.6	2391
	4.0	68.9	51.3	10.3	32.3	2394	64.6	58.3	13.3	30.6	2378
SEM ^2^		0.4	0.6	0.2	1.0	23	0.4	0.7	0.53	0.8	35
*p_s_*		0.71	0.86	0.94	0.55	0.70	0.61	0.58	0.93	0.57	0.99
*p_l_*		0.34	0.45	0.39	0.28	0.39	0.40	0.30	0.59	0.25	0.74
*p_s x l_*		0.12	0.10	0.15	0.11	0.10	0.15	0.38	0.39	0.40	0.20

^1^ SBO: Soybean oil; SLO: Soya Lecithin oil; L: Lard. ^2^ SEM = Standard Error of the Mean; n = 10. *p_s_*: probability of the source; *p_l_*: probability of the level; *p_s x l_*: probability of the interaction source x level.

**Table 9 animals-11-03389-t009:** Body and carcass chemical composition of animals at 63 d of age estimated by bioelectrical impedance (BIA) (experiment 2, individually housed) (N = 10).

		Whole Body Composition					Carcass Composition				
Fat ^1^ Source	Fat Level %	Water %	Protein %DM	Ash %DM	Fat %DM	Energy kJ/100 g DM	Water %	Protein %DM	Ash %DM	Fat %DM	Energy kJ/100 g DM
SBO	1.5	66.4	48.9	9.32	33.3	2467	62.7	54.9	11.8	33.6	2487
	4.0	66.5	48.8	9.18	33.1	2383	62.8	54.7	12.1	33.0	2466
FO	1.5	67.2	49.3	9.13	33.9	2491	63.0	55.1	12.4	31.9	2441
	4.0	67.3	49.8	9.35	33.1	2470	63.5	55.6	13.2	31.5	2393
PKO	1.5	67.1	49.2	9.32	33.2	2471	63.5	55.0	12.9	32.3	2418
	4.0	66.5	48.8	9.18	33.6	2466	63.1	54.9	12.3	33.1	2458
SEM ^2^		0.6	0.7	0.28	0.9	29	0.5	0.7	0.6	1.0	43
*p_s_*		0.71	0.37	0.59	0.97	0.98	0.92	0.52	0.72	0.41	0.32
*p_l_*		0.34	0.83	0.92	0.90	0.92	0.89	0.87	0.95	0.78	0.93
*p_s x l_*		0.15	0.77	0.81	0.82	0.79	0.81	0.65	0.86	0.58	0.81

^1^ SBO: Soybean oil; FO: Fish oil; PKO: Palm kernel oil. ^2^ SEM = Standard Error of the Mean; n = 10. *p_s_*: probability of the source; *p_l_*: probability of the level; *p_s x l_*: probability of the interaction source x level.

**Table 10 animals-11-03389-t010:** Effect of dietary fat source and fat inclusion level on estimated nitrogen and energy balance and excretion from 34 to 63 days of age (experiment 1, individually housed).

	Fat Source ^1^			Fat Level		rsd ^3^	*p_s_*	*p_l_*	*p_s x l_*
	SBO	SLO	L	1.5	4				
N	20	20	20	30	30				
Nitrogen Balance ^2^									
DNi ^2^, g/d	1.84	1.89	1.89	1.92	1.83	0.25	0.68	0.068	0.15
NR, g/d	0.673	0.674	0.694	0.671	0.690	0.109	0.64	0.37	0.58
NRE, %	36.5	35.8	36.7	34.9	37.8	0.03	0.45	<0.0001	0.10
Nitrogen Excretion, g/d									
Skin and organs	0.437	0.426	0.440	0.431	0.437	0.060	0.62	0.62	0.17
Feces	0.804	0.844	0.828	0.868	0.782	0.110	0.32	0.0001	0.12
Urine	0.736	0.788	0.762	0.822	0.702	0.144	0.33	<0.0001	0.11
Energy Balance									
DEi, kcal/d	305	310	311	301	316	41	0.83	0.15	0.10
RE, kcal/d	57.4	58.2	59.6	57.3	59.6	11.4	0.71	0.29	0.23
ERE, %	18.7	18.6	19.1	18.9	18.7	0.02	0.59	0.54	0.69
Energy Excretion, kcal/d									
Skin and organs	39.7	39.2	40.8	39.4	40.3	7.9	0.65	0.56	0.20
Feces	145	150	152	156	142	20	0.34	0.0004	0.12
Urine + heat	208	212	211	204	216	26	0.81	0.017	0.20

^1^ SBO: Soybean oil; SLO: Soya Lecithin oil; L: Lard. ^2^ DNi (g/d): Digestible Nitrogen Intake. NR (g/d): g N retained on the carcass; NRE: Efficiency of retention of Nitrogen; Skin and organs (g/d): (g N retained in the whole body—g N retained on the carcass); Feces (g/d): (N total intake—DNi. Urine (g/d): (DNi—g N retained on carcass—N excreted on skin and organs). DEi (kcal/d): Digestible Energy Intake. RE (kcal/d): Gross energy retained in the carcass; ERE: Efficiency of retention of Energy; Skin and organs (MJ): (GE retained in the whole body—GE retained in the carcass); Feces (MJ): (GEi—DEi). Urine + heat production (MJ): (Dei—GE retained in carcass—GE excreted in skin and organs). ^3^ rsd = residual standard deviation. *p_s_*: probability of the source; *p_l_*: probability of the level; *p_s x l_*: probability of the interaction source x level.

**Table 11 animals-11-03389-t011:** Effect of dietary fat source and fat inclusion level on estimated nitrogen and energy balance and excretion from 34 to 63 days of age (experiment 2, individually housed).

	Fat Source ^1^			Fat Level		rsd ^3^	*p_s_*	*p_l_*	*p_s x l_*
	SBO	FO	PKO	1.5	4				
N	20	20	20	30	30				
Nitrogen Balance ^2^									
DNi ^2^, g/d	1.93	1.78	1.91	1.95	1.79	0.24	0.10	0.014	0.74
NR, g/d	0.705	0.660	0.712	0.704	0.681	0.119	0.35	0.47	0.43
NRE, %	36.9	37.3	37.1	36.2	38.0	0.04	0.95	0.064	0.90
Nitrogen Excretion, g/d									
Skin and organs	0.431	0.415	0.439	0.442	0.415	0.059	0.47	0.10	0.96
Feces	0.829	0.755	0.832	0.865	0.745	0.103	0.38	<0.0001	0.79
Urine	0.781	0.698	0.761	0.799	0.694	0.157	0.23	0.014	0.93
Energy Balance									
DEi, kcal/d	303	279	301	295	293	38	0.20	0.875	0.72
RE, kcal/d	62.1	56.5	61.1	61.1	58.7	12.5	0.33	0.482	0.84
ERE, %	20.4	20.2	20.2	20.6	19.8	0.02	0.96	0.218	0.94
Energy Excretion, kcal/d									
Skin and organs	44.5	41.7	44.0	44.5	42.3	7.1	0.42	0.25	0.92
Feces	152	144	151	154	144	20	0.37	0.050	0.86
Urine + heat	196	181	196	182	182	25	0.19	0.66	0.59

^1^ SBO: Soybean oil; FO: Fish oil; PKO: Palm kernel oil. ^2^ DNi (g/d): Digestible Nitrogen Intake. NR (g/d): g N retained on the carcass; NRE: Efficiency of retention of Nitrogen; Skin and organs (g/d): (g N retained in the whole body—g N retained on the carcass); Feces (g/d): (N total intake—DNi. Urine (g/d): (DNi—g N retained on the carcass—N excreted on skin and organs). DEi (kcal/d): Digestible Energy Intake. RE (kcal/d): Gross energy retained in the carcass; ERE: Efficiency of retention of Energy; Skin and organs (MJ): (GE retained in the whole body—GE retained in the carcass); Feces (MJ): (GEi—DEi). Urine + heat production (MJ): (Dei—GE retained in carcass—GE excreted in skin and organs). ^3^ rsd = residual standard deviation. *p_s_*: probability of the source; *p_l_*: probability of the level; *p_s x l_*: probability of the interaction source x level.

## Data Availability

Data are not publicly available due to Trouw Nutrition confidentiality requirements.

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
