# Peer review of "Effect of Type and Dietary Fat Content on Rabbit Growing Performance and Nutrient Retention from 34 to 63 Days Old"

_animals, 2021, doi:10.3390/ani11123389_

Round 1

Reviewer 1 Report

 This Ms studied the impact of some fat sources added in different levels in growing rabbit feeds. In this study, it was found that the increase of dietary fat improved nitrogen efficiency and reduced nitrogen output, but the source of fat can affect performance and mortality. The Ms can be published after major revision considering the  comments reported in the revised copy attached and the following points:

1- The novelty of this study

2. The basis of choosing the level of fats 1.5 and 4%.

3. Reporting the experimental unit in the statistical section in the M&M section.

4. All abbreviations reported in the table must indicate the table footnotes? 

5. Plz indicate the no of replicates per each treatment per each experiment. Also please indicate the experimental unit?

6. Methione level can be moved above TSAA

7. Discussion on mortality must be improved due to unexpected results regarding the effect of fat types. 

8.  The level and dose of fat? must be added to the conclusion section 

9-References during 2018-2021 are absent and must be updated 

Reviewer 2 Report

GENERAL COMMENT:

I consider this work is within the scope of “Animals”. It contains information useful in a field in which available information is of great interest to improve knowledge on fat level and source on fattening rabbits. Overall, experimental design is correct and the sample size is good. However, I indicate several suggestions to be considered by the authors to improve the manuscript. I indicate these recommendations below and in a commented PDF file I have uploaded.

ABSTRACT:

The Abstract must begin with some background on the research aim and with reference to growing rabbits (at least, age/duration of the study period). Take into account that the Abstract is published independently of the article in the main bibliographic databases, and in its present form it does not permit to clearly understand that the experiment was carried out in fattening rabbits. In fact, no reference to rabbits has been done in the Abstract.

Line 19: Define with the full words the first time “DFI” initials appears in the Abstract; "...daily feed intake (DFI)..."

Line 20: Define with the full words the first time “FCR” initials appears in the Abstract: "...feed conversion ratio (FCR)..."

Line 23: Define with the full words the first time “BW” initials appears in the Abstract: "...body weight (BG)..."

Line 23: Define with the full words the first time “DWG” initials appears in the Abstract: "...daily weight gain (DWG)..."

ALONG THE ENTIRE MANUSCRIPT:

“k” letter from “kilo” must be written always as a lowercase letter.

INTRODUCTION:

Line 41: Remove hyphen where indicated.

Line 45: Duplicated [11] citation. Please remove one of them.

Line 53: Replace "animals" with "rabbits".

MATERIALS AND METHODS:

Line 143: Put in a more clear this reference: “RD 609/1999 nº4” (it is not found in the references section)

RESULTS:

This section is well organised.

DISCUSSION:

Line 325: Replace [32] with [33] where indicated.

CONCLUSIONS:

I recommend indicating that these conclusions are for growing rabbits.

REFERENCES SECTION:

In general terms, this section is well organised and adjusted to the style of the journal for references. However, some improvement is possible. For example:

I have indicated some typos in the commented version of the manuscript I have uploaded.

Several journal title abbreviations need to be corrected. I have indicated some of the in the commented version of the manuscript I have uploaded. For example:

Line 363: Use standard abbreviation: "Anim.", rather than “Animal”.

Line 382: Replace “BOE-A-2013-1337” with "BOE", volume umber of the official journal and page numbers of the law (first page-last page), similarly to a journal article.

TABLES:

Table 1: Correct “FiberFibereFiber”.

Table 1: Replace “Sepiolita” with “Sepiolite”

Tables 1 and 2, footnotes: Please revise that the numbers that must be typed as subscripts in the chemical formulas are typed correctly

Table 3: Type "1" as a superscript where indicated.

Table 3: Type: “w” where indicated.

Table 3: Remove accent marks where indicated.

Reviewer 3 Report

Manuscript animals-1459974, entitled “Effect of type and dietary fat inclusion on growing rabbit performance and nutrient retention from 34 to 63 days of age”

Recommendation:       The above paper is not suitable for publication in its present form.

General comment

The article provides useful information about the effects of type and level of dietary fat inclusion on growing rabbit performance and nutrient retention from 34 to 63 days of age. Although, the experiment is in general appropriately designed and implemented, there are some points that should be corrected or clarified.

Major comments

  • The presentation of the results needs further clarification. Table 7 is the same with Table 5 and in both tables fat sources of exp 1 and exp2 are confoundedly used. In Table 11, p-values for fat level and the interaction of fat source by fat level are the same with that of Table 10.
  • It is also important, especially in abstract, to state that fat types and levels were examined under individual and group housing of rabbits. At the same time, you should be specific; for example in L19-20, the results were found in individually or group housed rabbits?
  • Why were the other ingredients (apart from examined fat levels) different in 1.5 and 4.0 groups? As shown in Tables 1 and 2, diets were not isonitrogenous and isoenergetic (as stated in L320-321). Crude protein levels were lower in 1.5 vs 4.0% fat level groups both in exp. 1 and 2.
  • Energy and nitrogen carcass retention and excretion was measured in individually or group housed rabbits? Please add sample size in materials and methods (L120). Please add “individual housed rabbits” in Tables 8 and 9.
  • No mortality values are provided for individual housed rabbits
  • At the beginning of the discussion, authors try to compare individual and group housing. For example, FCR was not the same in the two systems (Exp. 1; 2.54 vs 2.43 in individually and group housed rabbits- No conclusion for Exp. 2, since Table 7 is incorrect). I think that authors reach to too general statements. A statistical analysis including housing type as an effect should be made for further conclusions.
  • L302-309: How are your results in agreement with that of previous studies? You found on average 36.5% NRE and the other authors 50.3, 41 and 51.5%. The same for ERE.

Minor points

Title: “Effect of type and levels of dietary fat…”

L16: “…mortality rates.”

L23: Similar mortality rates were observed for the lowest level of SBO

L37: “due” instead of “according”

L45-46: “In a previous study [11], mortality was reduced…”

L47: Control group received a caprylic-free diet?

L48: “Maertens et al.” instead of “in other work”

L50-52: Please rephrase (syntax errors)

L104: “recorded” instead of “controlled”

L108: Only in group housing

L160: No mortality values are shown

L161-162: “…was detected in the majority of the parameters recorded, independently…”

L178: “…higher than that with the lowest…”

L190-191: The values for BW49, DWG and DFI in SBO and PKO groups are not the same, so it is not correct to use only one percentage value.

L195: “compared to” instead of “with respect”

L199: Tendency for the total period

L202: “total one” instead of “whole”

Table 6 – DFI (49-63 days) – Fat source. Please use superscripts (P<0.05).

L208-213: Table 7 is incorrect, so these numbers cannot be checked.

L236-237: Inconsistency with Table 11

L265-266: “group and individually housed rabbits” instead of “animals allocated collectively and individually”

L272: “did not”

L273: “an improved” instead of “better”

L277: In group housing. Be precise

L278: “are not in agreement” instead of “disagree” “…increase in the mortality…”

L283: “…level, with a mortality rate of 3.5% on average.”

Author Response

Comments and edits of reviewer 3 were included directly in the main text.

Round 2

Reviewer 1 Report

No further comments

Author Response

We have revised English style and language. The reviewer 1 had no more comments.

Reviewer 3 Report

Although, the quality of article improved, some issues still remain, since authors did not correct them according to the comments of the first review round.

There are a lot of grammatical and syntax errors. For example, L36, 48, 58-61, 206-207, 232, 320, 327.

Title: "Effect of dietary fat type and levels on..."

L21-22: In Exp 1 or 2? If in Exp 1, DFI was significantly affected only in group-housed rabbits (in individually housed only between 34-49 days)

L26-28: In group housed rabbits?

L219, 238-240: The values for BW49, DWG and DFI in SBO and PKO groups are not the same, so it is not correct to use only one percentage value.

L228: It is a tendency not a significant effect

L322: It is not a tendency, it is a significant effect

L343-348: As discussed in the previous review round, these differences are not slight

L365-366: This statement is not correct, but misleading
